# Genome wide association analysis in a mouse advanced intercross line

Natalia M. Gonzales [1], Jungkyun Seo [2,3], Ana I. Hernandez Cordero[4], Celine L. St. Pierre[5], Jennifer S. Gregory[4], Margaret G. Distler[6], Mark Abney [1], Stefan Canzar[7], Arimantas Lionikas[4] & Abraham A. Palmer [8,9]

The LG/J x SM/J advanced intercross line of mice (LG x SM AIL) is a multigenerational outbred population. High minor allele frequencies, a simple genetic background, and the fully sequenced LG and SM genomes make it a powerful population for genome-wide association studies. Here we use 1,063 AIL mice to identify 126 significant associations for 50 traits relevant to human health and disease. We also identify thousands of cis- and trans-eQTLs in the hippocampus, striatum, and prefrontal cortex of ~200 mice. We replicate an association between locomotor activity and Csmd1, which we identified in an earlier generation of this AIL, and show that Csmd1 mutant mice recapitulate the locomotor phenotype. Our results demonstrate the utility of the LG x SM AIL as a mapping population, identify numerous novel associations, and shed light on the genetic architecture of mammalian behavior.

[1] Department of Human Genetics, University of Chicago, Chicago, IL 60637, USA. [2] Center for Genomic & Computational Biology, Duke University, Durham, NC 27708, USA. [3] Graduate Program in Computational Biology and Bioinformatics, Duke University, Durham, NC 27708, USA. [4] School of Medicine, Medical Sciences and Nutrition, College of Life Sciences and Medicine, University of Aberdeen, Aberdeen AB25 2ZD, UK. [5] Department of Genetics, Washington University School of Medicine, St. Louis, MO 63108, USA. [6] Department of Psychiatry and Biobehavioral Sciences, University of California Los Angeles, Los Angeles, CA 90095, USA. [7] Gene Center, Ludwig-Maximilians-Universität München, 81377 Munich, Germany. [8] Department of Psychiatry, University of California San Diego, La Jolla, CA 92093, USA. [9] Institute for Genomic Medicine, University of California San Diego, La Jolla, CA 92093, USA. Correspondence and requests for materials should be addressed to A.A.P. (email: aapalmer@ucsd.edu)

Genome-wide association studies (GWAS) have revolutionized psychiatric genetics; however, they have also presented numerous challenges. Some of these challenges can be addressed by using model organisms. For example, human GWAS are confounded by environmental variables, such as childhood trauma, which can reduce power to detect genetic associations. In model organisms, environmental variables can be carefully controlled. Furthermore, it has become clear that phenotypic variation in humans is due to numerous common and rare variants of small effect. In model organisms, genetic diversity can be controlled such that all variants are common. In addition, allelic effect sizes in model organisms are dramatically larger than in humans[1,2]. Because the majority of associated loci are in noncoding regions, expression quantitative trait loci (eQTLs) are useful for elucidating underlying molecular mechanisms[3,4]. However, it remains challenging to obtain large, high-quality samples of human tissue, particularly from the brain. In contrast, tissue for gene expression studies can be collected from model organisms under optimal conditions. Finally, the genomes of model organisms can be edited to assess the functional consequences of specific mutations.

Unlike classical $F_2$ crosses, outbred animals provide improved mapping resolution for GWAS. This is because outbred populations have higher levels of recombination, meaning only markers very close to the causal allele will be associated with the phenotype. However, there is a necessary tradeoff between mapping resolution and statistical power, and this is further aggravated when the causal allele is rare, which is sometimes the case in commercially outbred mice[5,6]. In an effort to combine the resolution afforded by an outbred population with the power of an $F_2$ cross, we performed GWAS using an advanced intercross line (AIL) of mice. Originally proposed by Darvasi and Soller[7], AILs are the simplest possible outbred population; they are produced by intercrossing two inbred strains beyond the $F_2$ generation. Because each inbred strain contributes equally to an AIL, all variants are common. This avoids the loss of power that results from rare alleles and simplifies phasing and imputation. Each successive generation of intercrossing further degrades linkage disequilibrium (LD) between adjacent markers, which improves mapping resolution relative to classical inbred crosses such as $F_2$s.

An AIL derived from the LG/J (LG) and SM/J (SM) inbred strains was initiated by Dr. James Cheverud at Washington University in St. Louis[8]. In 2006, we established an independent AIL colony with mice from generation 33 at the University of Chicago (G33; Jmc:LG,SM-G33). Because the LG and SM founder strains were independently selected for large and small body size, this AIL has frequently been used to study body weight, musculoskeletal, and other metabolic traits[9–12]. LG and SM also exhibit a variety of behavioral differences, and previous studies from our lab have used the LG × SM AIL to fine-map behavioral associations identified in $F_2$ crosses between LG and SM[13–15].

We recently used commercially available outbred CFW mice to perform a GWAS[5]. In this paper, we apply a similar approach to AIL mice, taking advantage of the more favorable allele frequencies and the ability to impute founder haplotypes. Using mice from AIL generations 50–56 (G50–56; Aap:LG,SM-G50–56), we investigate novel behavioral traits, including conditioned place preference for methamphetamine (CPP), as well as other biomedically important traits including locomotor activity following vehicle and methamphetamine administration, which have been extensively studied for their relevance to drug abuse[16], and prepulse inhibition (PPI), which has been studied for several decades as an endophenotype for schizophrenia[17]. We also examine body weight and a constellation of musculoskeletal phenotypes relevant to exercise physiology, which are known to be heritable in this AIL[10,11,18]. In total, we report 52 associations for 33 behavioral traits and 74 associations for 17 physiological traits. We use RNA-sequencing (RNA-seq) to measure gene expression in three brain regions of ~200 mice and identify thousands of cis- and trans-eQTLs, which we use to identify quantitative trait genes (QTGs). Finally, we use a mutant mouse line to validate one of our strongest candidate QTGs. Our work demonstrates that the LG × SM AIL is a powerful tool for genetic analysis and provides a methodological framework for GWAS in multigenerational outbred populations. The numerous associations described in this work benefit the complex trait community by shedding light on the genetic architecture of complex traits in a system with only two founders. Although we present only a subset of our findings in detail, AIL genotype, phenotype and gene expression data, complete with a suite of tools for performing GWAS and related analyses, are publicly available on GeneNetwork.org[19].

## Results

**Genotyping by sequencing.** We used genotyping by sequencing (GBS) to genotype 1063 of the 1123 mice that were phenotyped (60 were not successfully genotyped for technical reasons described in the Supplementary Methods). After quality control, GBS yielded 38,238 high-quality autosomal SNPs. Twenty-four AIL mice were also genotyped using the Giga Mouse Universal Genotyping Array[20] (GigaMUGA), which yielded only 24,934 polymorphic markers (Supplementary Figure 1). LG and SM have been re-sequenced[21], which allowed us to use the GBS genotypes to impute an additional 4.3 million SNPs (Fig. 1a). Consistent with the expectation for an AIL, the average minor allele frequency (MAF) was high (Fig. 1b). We also observed that the decay of LD, which is critical to mapping resolution, has increased since the 34th generation (Fig. 1c).

**LOCO-LMM effectively reduces the type II error rate.** Linear mixed models (LMMs) are now commonly used to perform GWAS in populations that include close relatives because they can incorporate a genetic relationship matrix (GRM) that models the covariance of genotypes and phenotypes in samples of related individuals[22]. If SNP data are used to obtain the GRM, this can lead to an inflation of the type II error rate due to proximal contamination[23,24]. We have proposed using a leave-one-chromosome-out LMM (LOCO-LMM) to address this issue[23]. To demonstrate the appropriateness of a LOCO-LMM, we performed a GWAS for albinism, which is known to be a recessive Mendelian trait caused by the $Tyr^c$ allele, using all three approaches: a simple linear model, an LMM, and a LOCO-LMM (Fig. 2). GWAS using a LOCO-LMM for albinism yielded an association on chromosome 7, which contains the $Tyr^c$ allele (Fig. 2a). As expected, a quantile-quantile plot showed that p-values from a genome-wide scan using a linear model, which does not account for relatedness, appeared highly inflated (Fig. 2b). This inflation was greatly reduced by fitting a standard LMM, which included SNPs from chromosome 7 in both the fixed and random terms (Fig. 2c). The LOCO-LMM, which does not include SNPs from the chromosome being tested in the GRM, showed an intermediate level of inflation (Fig. 2d). Was the inflation observed in Fig. 2b–d due to true signal, or uncontrolled population structure? To address this question, we repeated these analyses after excluding SNPs on chromosome 7 from the fixed effect (Fig. 2e–g). Even in the absence of the causal locus, the simple linear model showed substantial inflation, which can only be explained by population structure (Fig. 2e). The standard LMM appeared overly conservative, which we attributed to proximal contamination (Fig. 2f). The LOCO-LMM showed no inflation, consistent with the absence of $Tyr^c$ and linked SNPs in

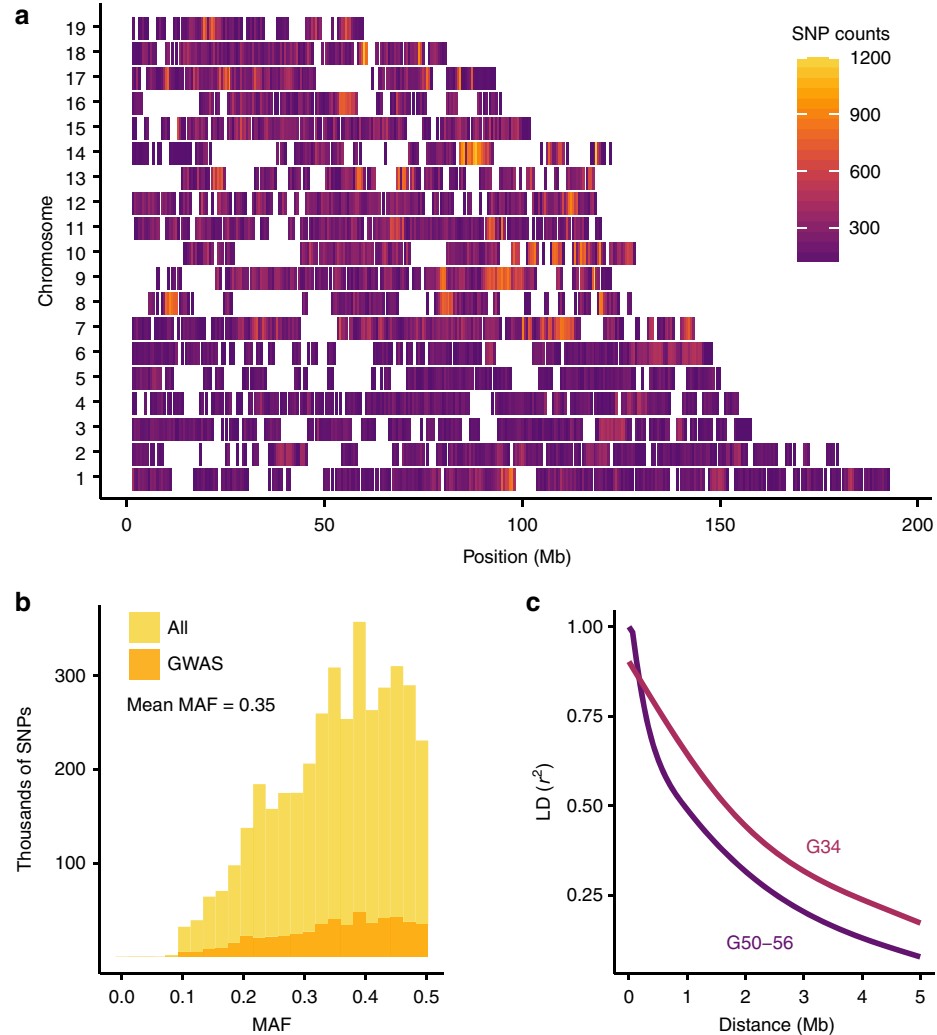

**Fig. 1** Variants, MAFs and LD decay in the LG × SM AIL. Imputation provided 4.3 million SNPs. Filtering for LD ($r^2 \geq 0.95$), MAF < 0.1, and HWE ($p \leq 7.62 \times 10^{-6}$, Chi-squared test) resulted in 523,028 SNPs for GWAS. **a** SNP distribution and density of GWAS SNPs are plotted in 500 kb windows for each chromosome. As shown in Supplementary Figure 1, regions with low SNP density correspond to regions predicted to be nearly identical by descent in LG and SM[21]. **b** MAF distributions are shown for 4.3 million imputed SNPs (gold; unfiltered) and for the 523,028 SNPs used for GWAS (orange; filtered). Mean MAF is the same in both SNP sets. **c** Comparison of LD decay in G50–56 (dark purple) and G34 (light purple) of the LG × SM AIL. Each curve was plotted using the 95th percentile of $r^2$ values for SNPs spaced up to 5 Mb apart

the fixed effect (Fig. 2g). These results demonstrate the appropriateness of a LOCO-LMM.

**Genetic architecture of complex traits in the LG × SM AIL.** We used an LD-pruned set of 523,028 autosomal SNPs genotyped in 1,063 mice from LG × SM G50–56 to perform GWAS for 120 traits using a LOCO-LMM (Fig. 3a). Although our primary interest was in behavior, we also measured a number of physiological traits that we knew to be heritable in this AIL. We used permutation to define a significance threshold of $p = 8.06 \times 10^{-6}$ at $\alpha = 0.05$ (all $p$-values describing GWAS or eQTL results were obtained using the likelihood ratio test in GEMMA). We did not use a Bonferroni-corrected significance threshold because we have previously shown that permutation adequately controls type I and II error rates[25]. We identified 52 loci associated with 33 behavioral traits and 74 loci associated with 17 physiological traits (Fig. 3a, Supplementary Data 1; Supplementary Figure 2).

In populations of related individuals, heritability ($h^2$) estimates may differ depending on whether pedigree data or genetic data is used to calculate the GRM[15,22,26]. Pedigree data captures shared

environmental effects and epigenetic effects in addition to additive genetic effects. We used GEMMA[27] to estimate $h^2$ using a GRM calculated from the AIL pedigree ($\hat{h}^2_{Ped}$) and compared the results to $h^2$ estimates obtained using a GRM calculated from 523,028 SNPs ($\hat{h}^2_{SNP}$). Both methods produced higher estimates of $h^2$ for physiological traits than for behavioral traits (Supplementary Data 2), which is consistent with findings in other rodent GWAS[5,6,28]. For behavioral traits (excluding CPP, locomotor sensitization, and habituation to startle, which were not found to have a genetic component), mean $\hat{h}^2_{Ped}$ was 0.172 (s.e.m. = 0.066) and mean $\hat{h}^2_{SNP}$ was 0.168 (s.e.m. = 0.038). For physiological traits, $\hat{h}^2_{Ped}$ was 0.453 (s.e.m. = 0.088) and $\hat{h}^2_{SNP}$ was 0.355 (s.e.m. = 0.045). The lower standard error for $\hat{h}^2_{SNP}$ indicates that it is a more powerful estimator of $h^2$ than $\hat{h}^2_{Ped}$. $\hat{h}^2_{SNP}$ is displayed for a subset of traits in Fig. 3b; a complete list of $h^2$ estimates and their standard errors are reported in Supplementary Data 2. In general, traits with higher heritabilities yielded more associations (Supplementary Figure 2). However, there was no relationship

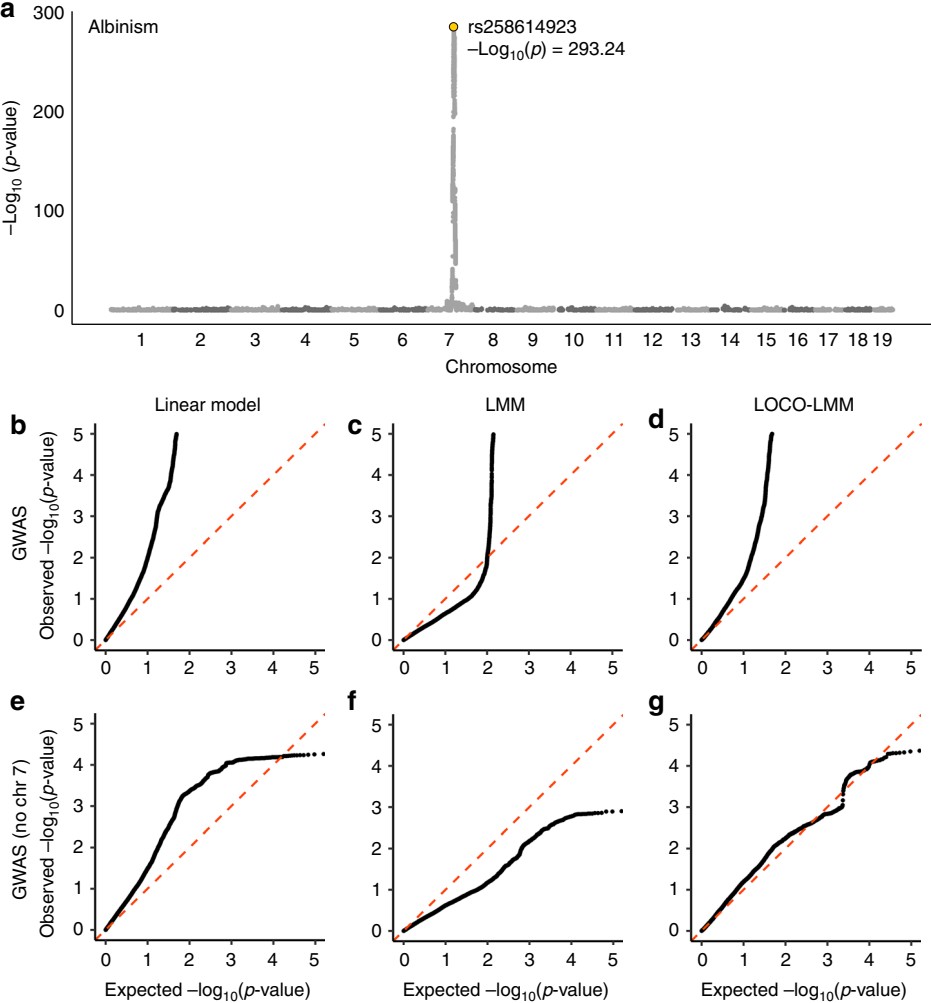

**Fig. 2** GWAS for albinism verifies that the LOCO-LMM effectively controls type I and type II error. We conducted a GWAS for albinism, a Mendelian trait caused by the *Tyr^c* locus on mouse chromosome 7, using three models: a linear model, an LMM, and a LOCO-LMM. We also repeated each scan after excluding SNPs on chromosome 7. A Manhattan plot of results from the LOCO-LMM is shown in (**a**). Quantile-quantile plots of expected vs. observed *p*-values are shown for (**b**) a simple linear model that does not account for relatedness; **c** a standard LMM that includes all GWAS SNPs in the GRM (i.e., the random effect); and **d** a LOCO-LMM whose GRM excludes SNPs located on the chromosome being tested. Plots **e–g** show results after excluding chromosome 7 from the GWAS

between MAF and the percent variance explained by individual loci (Supplementary Figure 3).

We expected many of the traits measured in this study to be correlated with one another (as explained in the Methods and Supplementary Methods). Thus, we calculated Pearson correlation coefficients for all pairwise combinations of traits measured in this study (Supplementary Figure 4). We also used GEMMA to estimate the proportion of genetic and environmental variance shared by each pair of traits (Supplementary Figure 5, Supplementary Data 3). As expected, behavioral trait correlations were highest for traits measured on the same day, and correlations among body size traits were also high, even for traits measured at different time points. Trait pairs with high heritabilities shared a larger proportion of genetic variance with one another than with traits that had low heritabilities. Very little shared genetic variance was observed for behavioral-physiological trait pairs, consistent with their generally low trait correlations.

**Identification of eQTLs using RNA-seq**. For a subset of phenotyped and genotyped mice, we used RNA-seq to measure gene expression in the hippocampus (HIP; $n = 208$), prefrontal cortex

(PFC; $n = 185$) and striatum (STR; $n = 169$) (Supplementary Figure 6). We used this data to map local eQTLs (located up to 1 Mb from the start and end of each gene), which we refer to as *cis*-eQTLs. We identified 2902 *cis*-eQTLs in HIP, 2125 *cis*-eQTLs in PFC and 2054 *cis*-eQTLs in STR; 1087 *cis*-eQTLs were significant in all three tissues (FDR < 0.05; Supplementary Data 4, Supplementary Figures 6–7).

We also mapped distal eQTLs (located on a separate chromosome from the gene tested), which we refer to as *trans*-eQTLs. We identified 723 HIP *trans*-eQTLs ($p < 9.01 \times 10^{-6}$), 626 PFC *trans*-eQTLs ($p < 1.04 \times 10^{-5}$) and 653 STR *trans*-eQTLs ($p < 8.68 \times 10^{-6}$) at a genome-wide significance threshold of $\alpha = 0.05$ after testing 49,642 genes across the three tissues (Supplementary Figures 6–7; Supplementary Data 5). Because our permutation corrects for all SNPs tested for a single gene, but not all for all genes and tissues tested, we would expect 5% of tests to be false positives. Quantile-quantile plots that included all SNPs and all genes in each tissue revealed an excess of low *p*-values, suggesting the presence of true positive results (Supplementary Figure 8).

Previous studies in model organisms have identified *trans*-eQTLs that regulate the expression of many genes[4,29,30]; we refer

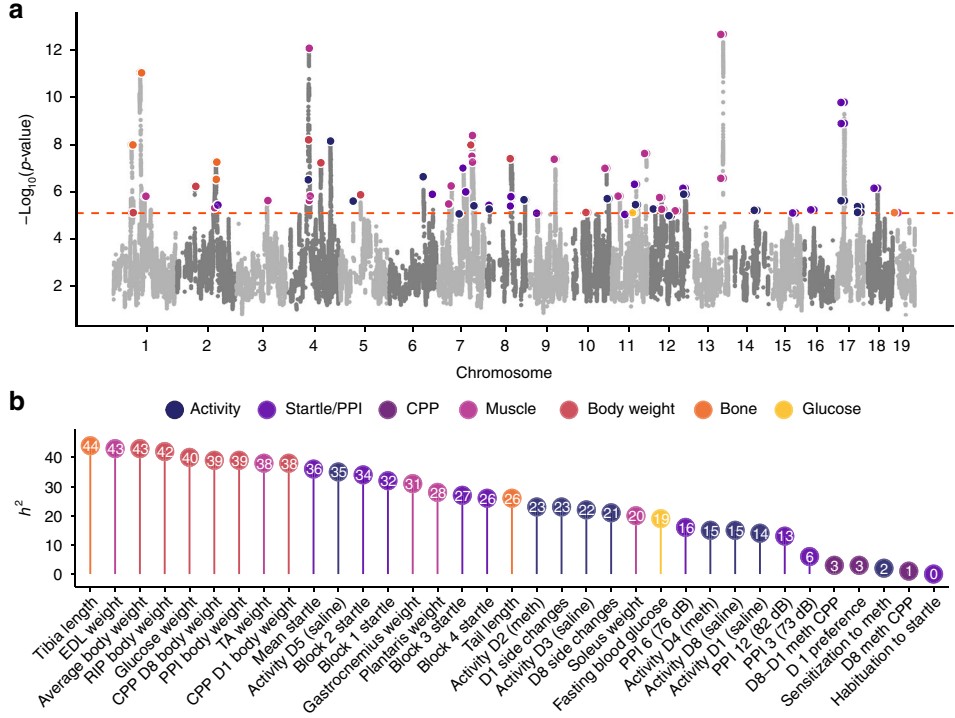

**Fig. 3** Manhattan plot and heritability for 120 traits measured in the LG × SM AIL. We identified 126 loci for behavioral and physiological traits using 1063 mice from G50–56 of the LG × SM AIL. A Manhattan plot of GWAS results is shown in (**a**). Associations for related traits are grouped by color. For clarity, related traits that mapped to the same locus (Supplementary Data 1) are highlighted only once. The dashed line indicates a permutation-derived significance threshold of $-\log_{10}(p) = 5.09$ ($p = 8.06 \times 10^{-6}$; $\alpha = 0.05$; likelihood ratio test). **b** For a representative subset of traits, SNP heritability estimates (percent trait variance explained by 523,028 GWAS SNPs) are shown. Precise estimates of heritability with standard error are provided for all traits in Supplementary Data 2

to these as master eQTLs (others have called them *trans*-bands). We identified several master eQTLs, including one on chromosome 12 (70.19–73.72 Mb) that was associated with the expression of 97 genes distributed throughout the genome (Supplementary Figure 9; Supplementary Data 5). This locus was present in HIP, but not in PFC or STR.

**Integration of GWAS and eQTL results**. The number of significant associations precludes a detailed discussion of each locus. Instead, we have chosen to present several examples that show how we used various layers of complimentary data to parse among the genes within implicated loci. Manhattan plots for all traits are included in Supplementary Figure 2. In addition, all of our data are available on GeneNetwork.org[19], which is a website that provides statistical tools and an interactive graphical user interface, allowing the user to replicate our results and explore additional results not presented in this paper.

Four loci associated with locomotor behavior mapped to the same region on chromosome 17 (Supplementary Data 1; Supplementary Figure 2). The narrowest association was for side changes between 15–20 min on day 1 (D1) of the CPP test, after mice received an injection of vehicle ($p = 3.60 \times 10^{-6}$). Other genome-wide significant associations included total D1 side changes (0–30 min), distance traveled on D1 (0–5 min), and distance traveled after an injection of methamphetamine on D2 (15–20 min; Supplementary Data 1). The implicated locus contains a single gene, *Crim1* (cysteine-rich transmembrane BMP regulator 1), which had a significant *cis*-eQTL in HIP. Although *Crim1* may appear to be the best candidate to explain the associations with locomotor behavior, two nearby genes, *Qpct* (glutaminyl-peptide cyclotransferase) and *Vit* (vitrin), though

physically located outside of the locus, also had *cis*-eQTLs within the region associated with locomotor behavior (Supplementary Data 4). The top SNP at this locus (rs108572120) is also a *trans*-eQTL for *Zfas1* expression in STR. *Zfas1* is a noncoding RNA on chromosome 2 of unknown significance. We therefore consider all four genes candidates for mediating the association between this locus and locomotor behavior.

One of the most significant loci we identified was associated with startle response ($p = 5.28 \times 10^{-10}$; Fig. 3; Supplementary Figure 2). The startle response is a motor reflex that is used to assess neurobiological traits related to behavioral plasticity and sensorimotor processing[17,31]. This result replicated a previous association with startle response from our prior study using G34 AIL mice[14]. We examined GWAS p-values for the most significantly associated startle SNP (rs33094557) across all traits that we measured in this study (we refer to this as a phenome-wide association analysis, or PheWAS[32]), which revealed that this region pleiotropically affected multiple other traits, including locomotor activity following saline and methamphetamine administration (Supplementary Figure 10). This region was also implicated in conditioned fear and anxiety in prior studies of G34 mice[15], demonstrating that it has extensive pleiotropic effects on behavior. Because the association with startle identifies a relatively large haplotype that included over 25 genes with eQTLs, the causal gene(s) is not clear, and we are not certain whether the pleiotropic effects are due to one or several genes in this interval. In this case, the mapping resolution we obtained was insufficient to address these questions.

We also identified a 0.49-Mb locus on chromosome 8 that was associated with the locomotor activity (Fig. 4a; Supplementary Data 1); this region was nominally associated with PPI and multiple other locomotor traits (Fig. 4b). The region identified in

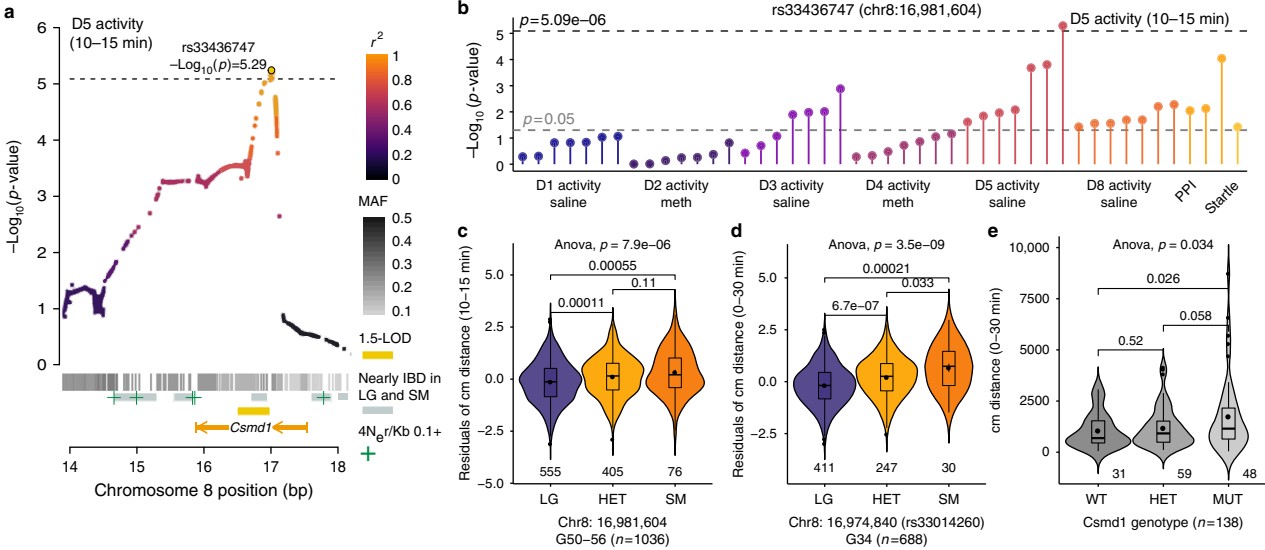

**Fig. 4** Replication of an association between *Csmd1* and locomotor activity. **a** Regional plot drawn from the full set of 4.3 million unpruned SNPs showing the association between rs33436747 and D5 activity levels (12,058 SNPs are plotted for this region). The location of *Csmd1*, 1.5-LOD interval (gold bar), areas of elevated recombination[79] (green plus symbols), regions predicted to be nearly IBD between LG and SM[21] (gray bars), and SNP MAFs (gray heatmap) are indicated. Points are colored by LD ($r^2$) with rs33436747 The dashed line indicates a significance threshold of $-\log_{10}(p) = 5.09$ ($\alpha = 0.05$; likelihood ratio test). **b** PheWAS plot of associations between rs33436747 and other behavioral traits measured in G50–56 mice. Violin plots in (**c–e**) display the interquartile range with the median plotted as a horizontal line and the mean plotted as a point. *P*-values from ANOVA and two-sided *t*-tests are shown for each comparison. **c** Violin plot of quantile-normalized residuals of locomotor activity at the *Csmd1* locus are plotted for G50–56 mice (ANOVA $p = 7.85e$ $-06$; $F = 11.89$; df = 2, 1033). For each comparison, LG vs. SM $p = 5.52e-04$, $t = -3.58$, df = 94.34; LG vs. HET $p = 1.11e-04$, $t = -3.88$, df = 890.44; SM vs. HET $p = 0.108$, $t = 1.62$, df = 99.56. **d** Violin plot of quantile-normalized residuals of locomotor activity at the *Csmd1* locus for G34 mice[13] (ANOVA $p =$ $3.51e-09$; $F = -20.03$; df = 2, 685). For each comparison, LG vs. SM $p = 2.11e-04$, $t = -4.17$, df = 32.45; LG vs. HET $p = 6.72e-07$, $t = -5.03$, df = 521.27; SM vs. HET $p = 0.033$, $t = 2.22$, df = 34.73. rs33436747 was not genotyped in G34; therefore, we plotted activity by genotype at the nearest SNP (rs33014260); 6764 bp upstream of rs33436747. **e** Violin plot of locomotor activity data (distance traveled in 0–30 min) for *Csmd1* mutant (MUT) mice (ANOVA $p = 0.034$; $F = 3.46$; df = 2, 135). For each comparison, WT vs. MUT $p = 0.026$, $t = -2.27$, df = 71.03; WT vs. HET $p = 0.519$, $t = -0.649$, df = 68.20; MUT vs. HET $p = 0.058$, $t = -1.93$, df = 67.62. Data plotted in (**c–e**) is provided as a Source Data file

the present study (Fig. 4a–c) replicates a finding from our previous study using G34 AIL mice[13] (Fig. 4d). In both cases, the SM allele conferred increased activity (Fig. 4c, d) and the implicated locus contained only one gene: *Csmd1* (CUB and sushi multiple domains 1; Fig. 4b; Supplementary Data 2); furthermore, the only *cis*-eQTL that mapped to this region was for *Csmd1* (Supplementary Figure 11). We obtained mice in which the first exon of *Csmd1* was deleted to test the hypothesis that *Csmd1* is the QTG for this locus. *Csmd1* mutant mice exhibited increased activity compared to heterozygous and wild-type mice (Fig. 4e), similar to the SM allele. Taken together, these data strongly suggest that *Csmd1* is the causal gene.

We identified seven overlapping loci for locomotor activity on chromosome 4 (Supplementary Data 1; Supplementary Figure 2). The strongest locus (D5 activity, 0–30 min; $p = 6.75 \times 10^{-9}$) spanned 2.31 Mb and completely encompassed the narrowest locus, which spanned 0.74 Mb (D5 activity, 25–30 min; $p = 4.66 \times 10^{-8}$); therefore, we focused on the smaller region. *Oprd1* (opioid receptor delta 1) had a *cis*-eQTL in all three brain regions; the SM allele conferred an increase in locomotor activity and was associated with decreased expression of *Oprd1*. *Oprd1* knockout mice have been reported to display increased activity relative to wild-type mice[33], suggesting that differential expression of *Oprd1* could explain the locomotor effect at this locus. However, we note that there are several other genes and eQTLs within this locus that could also contribute to its behavioral effects.

Finally, we identified an association with D1 locomotor behavior on chromosome 6 at rs108610974, which is located in an intron of *Itpr1* (inositol 1,4,5-trisphosphate receptor type 1; Supplementary Figure 12). This locus contained three *cis*-eQTLs

and seven *trans*-eQTLs (Supplementary Figure 12). One of the *trans*-eQTL genes targeted by the locus (*Capn5*; calpain 5) was most strongly associated with rs108610974, and may be the QTG (Supplementary Data 5). These results illustrate how knowledge of both *cis*- and *trans*-eQTLs informed our search for QTGs.

**Pleiotropic effects on physiological traits.** Because LG and SM were created by selective breeding for large and small body size, this AIL is expected to segregate numerous body size alleles[9,10]. We measured body weight at ten timepoints throughout development and identified 46 associations, many of which converged at the same loci (Supplementary Data 1, Supplementary Figure 2). For example, eight body weight timepoints were associated with a locus on chromosome 2. Counter to expectations, the LG allele at this locus was associated with smaller body mass (Supplementary Data 1; Supplementary Figure 2, Supplementary Figure 13). The narrowest region spanned 0.08 Mb, and while it did not contain any genes, it did contain a *cis*-eQTL for *Nr4a2* (nuclear receptor subfamily 4, group A, member 2) in PFC. Mice lacking *Nr4a2* in midbrain dopamine neurons exhibit a 40% reduction in body weight[34]. Consistent with this, the LG allele was associated with decreased expression of *Nr4a2*. Taken together, these data strongly implicate *Nr4a2* as the QTG for this locus.

LG and SM also exhibit differences in fat and muscle weight, bone density, and other morphometric traits[9–11,35]. Therefore, we measured the weight of five hindlimb muscles, tibia length, and tail length (Methods). We identified 22 associations for muscle weight and five for bone length (Supplementary Data 1). Six muscle weight loci and one tibia length locus overlapped with

associations for body weight. Since body weight, muscle weight, and bone length are interdependent traits, it is difficult to know whether this overlap is due to these traits being correlated (Supplementary Figures 4–5, Supplementary Data 3) or if the causal SNP(s) affect these traits through independent mechanisms. Thus, we use the term pleiotropy to describe these loci instead of claiming a causal relationship. For example, associations for tibialis anterior (TA), gastrocnemius, and plantaris weight overlapped a region on chromosome 7 that was associated with all body weight timepoints that we measured (Supplementary Figure 2, Supplementary Figure 14, Supplementary Data 1). Although the most significant SNP associated with muscle weight was ~5 Mb downstream of the top body weight SNP, the LG allele was associated with greater weight at both loci (Supplementary Data 1). For eight out of ten body weight timepoints, the most significant association fell within *Tpp1* (tripeptidyl peptidase 1), which was a *cis*-eQTL gene in all tissues and a *trans*-eQTL gene targeted by the master HIP eQTL on chromosome 12 (Supplementary Figure 9). To our knowledge, *Tpp1* has not been shown to affect body size in mice or humans; however, four other *cis*-eQTL genes in the region have been associated with human body mass index (*Rpl27a*, *Stk33*, *Trim66*, and *Tub*)[36,37]. Dysfunction of *Tub* (tubby bipartite transcription factor) causes late-onset obesity in mice, perhaps due to *Tub*'s role in insulin signaling[38]. In addition, several *trans*-eQTL genes map to this interval, including *Gnb1* (G protein subunit beta 1), which forms a complex with *Tub*[39]. Another *trans*-eQTL gene associated with this interval, *Crebbp* (CREB binding protein), has been associated with juvenile obesity in human GWAS[34].

The strongest association we identified in this study was for EDL weight ($p = 2.03 \times 10^{-13}$) on chromosome 13 (Fig. 3a, Supplementary Data 1; Supplementary Figure 15). An association with gastrocnemius weight provided additional support for the region ($p = 2.56 \times 10^{-7}$; Supplementary Figure 2, Supplementary Figure 15) and in both cases, the SM allele was associated with increased muscle mass. Each locus spanned less than 0.5 Mb and was flanked by regions of low polymorphism between LG and SM (Supplementary Figure 15, Supplementary Data 1). A *cis*-eQTL gene within this region, *Nln* (neurolysin), is differentially expressed in LG and SM soleus muscle[12,18], with LG exhibiting greater expression. *Nln* has been shown to play a role in mouse skeletal muscle[40].

Finally, we identified an association with EDL, plantaris, and TA weight at another locus on chromosome 4 (Supplementary Data 1; Supplementary Figure 16). In all cases, the LG allele was associated with greater muscle weight. The loci for EDL and plantaris spanned ~0.5 Mb, defining a region that contained six genes (Supplementary Data 1). The top SNPs for EDL (rs239008301; $p = 7.88 \times 10^{-13}$) and plantaris (rs246489756; $p = 2.25 \times 10^{-6}$) were located in an intron of *Astn2* (astrotactin 2), which is differentially expressed in LG and SM soleus[12]. SM, which exhibits lower expression of *Astn2* in soleus relative to LG[12], has a 16 bp insertion in an enhancer region 6.6 kb upstream of *Astn2* (ENSMUSR00000192783)[21]. Two other genes in this region have been associated with muscle or bone phenotypes traits in the mouse: *Tlr4* (toll-like receptor 4), which harbors one synonymous coding mutation on the SM background (rs13489095) and *Trim32* (tripartite motif-containing 32), which contains no coding polymorphisms between the LG and SM strains.

## Discussion

Crosses among well-characterized inbred strains are a mainstay of model organism genetics. However, $F_2$ and similar crosses provide poor mapping resolution because the ancestral chromosomes

persist as extremely long haplotypes[2,22]. To address this limitation, we and others have used various multigenerational intercrosses, including AILs. Using 1063 male and female mice from LG × SM G50–56, we confirmed that most variants in the LG × SM AIL had high frequencies and that LD has continued to decay between G34 and G50–56 (Fig. 1). We identified 126 loci for traits selected for their relevance to human psychiatric and metabolic diseases[9,16,17] (Fig. 3; Supplementary Data 1; Supplementary Figure 2). These results implicated several specific genes that are corroborated by extant human and mouse genetic data. In particular, we replicated a locus on chromosome 8 that was associated with locomotor activity in the G34 study[13] (Fig. 4). We showed that the chromosome 8 locus contained a *cis*-eQTL for *Csmd1* (Supplementary Figure 11), which is the only annotated gene within that locus. Finally, we showed that a genetically engineered *Csmd1* mutant mouse recapitulates the locomotor phenotype, strongly suggesting that *Csmd1* is the QTG.

Our previous work with this AIL used mice from LG × SM G34 to fine-map loci identified in an $F_2$ cross between LG and SM; however, the mapping resolution in those studies was limited by a lower number of generations, fewer markers (~3000 SNPs), a smaller sample size, and a higher proportion of first-degree relatives[10,12–14,18]. In the present study, we addressed these limitations. The number of SNPs used in our prior studies was increased by several orders of magnitude by using GBS[41,42] followed by imputation from the sequenced founders[21]. Unlike our previous studies using the G34 AIL, in this study we did not use AIL mice to fine map loci identified in an $F_2$ cross, but instead used them as both a discovery and fine mapping population.

The strategies we used to perform GWAS in the LG × SM AIL were also informed by recent GWAS using outbred CFW mice[5,6]. Although our approach to the AIL was similar to the CFW studies, certain practical considerations, along with the AIL's simpler genetic background, affected the design of this study and its outcomes in important ways. For example, non-sibling CFW mice can be obtained from a commercial vendor, which avoids the expenses of maintaining an AIL colony and reduces the complicating effects that can occur when close relatives are used in GWAS. However, haplotype data from the CFW founders are not available, and many CFW alleles exist at low frequencies, limiting power and introducing genetic noise[5,6]. In contrast, the LG and SM founder strains have been fully sequenced[21] and AIL SNPs have high MAFs, which simplifies imputation of SNPs and founder haplotypes and enhances statistical power[22]. Parker et al. demonstrated that GBS is a cost-effective strategy for genotyping CFW mice[5]; however, in the present study, we took advantage of the fact that in the LG × SM AIL, all alleles that are identical by state are also identical by descent. This allowed us to use imputation to obtain 4.3 million SNPs despite using only about half the sequencing depth that was necessary for CFW mice (Fig. 1a). Even before imputation, GBS yielded nearly 50% more informative SNPs compared to the best available SNP genotyping chip[20] at about half the cost (Supplementary Figure 1). Thus, we have shown that the use of GBS in a population with sequenced founders is even more powerful than the analogous approach in CFW mice.

The primary goal of this study was to identify the genes that are responsible for the loci implicated in behavioral and physiological traits. We were particularly interested in uncovering genetic factors that influence CPP, which is a well-validated measure of the reinforcing effects of drugs[43]. However, the heritability of CPP in LG × SM G50–56 was not significantly different from zero (Fig. 3b). This was unexpected, since panels of inbred strains and genetically engineered mutant alleles have been shown to exhibit heritable differences in CPP[43–45]. However, the lack of heritability of CPP was partially consistent with our prior study, which used a

higher dose of methamphetamine (2 vs. the 1 mg kg$^{-1}$ used in the present study)[46]. We conclude that the low heritability of CPP likely reflects a lack of relevant genetic variation in this specific population. It is possible that even lower doses of methamphetamine, which might fall on the ascending portion of the dose-response function, would have resulted in higher heritability. Responses to other drugs or different CPP methodology may also have resulted in higher heritability. It is also possible that testing for non-additive genetic effects would have increased the heritability of CPP.

In general, trait heritability in the LG × SM AIL (Fig. 3b, Supplementary Data 2) was lower than what has been reported in studies that use panels of inbred mouse strains to estimate heritability[47–49]. This was expected because environmental effects can be shared within but not between inbred strains, which will produce higher estimates of heritability[26]. In addition, epistatic interactions can contribute to heritability in inbred panels, which leads to higher estimates of heritability[50]. These observations share many similarities with the higher heritabilities obtained using twins compared to chip heritabilities in human studies[26].

A subset of traits was measured in both this AIL and our prior CFW study (PPI, startle, and body size). Heritability estimates were very similar between those two populations[5]. Despite this, we identified many more genome-wide significant loci in the current study (126 associations in the AIL compared to 17 in CFW at α = 0.05). The greater number of significant associations generated in this study may be due to enhanced power that comes with the more favorable distribution of MAFs in the AIL (Supplementary Figure 3). Differences in genetic background and the rate of LD decay, which also affect power, may also have contributed to these observations and could account for the lack of overlap among loci associated with PPI, startle and body size in the two populations.

Our ability to identify QTGs was critically dependent on mapping resolution in the LG × SM AIL (Fig. 1c). However, proximity of a gene to the associated SNP is insufficient to establish causality[4]. Therefore, we used RNA-seq to quantify mRNA abundance in three brain regions that are strongly implicated in the behavioral traits that we measured: HIP, PFC, and STR. We used these data to identify 7081 cis-eQTLs (FDR < 0.05) and 2002 trans-eQTLs (α = 0.05 for a single gene test) (Supplementary Figures 6–9, Supplementary Data 4–5). In a few cases, loci contained only a single eQTL, but in most cases, multiple cis-eQTLs and trans-eQTLs mapped to the implicated loci. This highlights an important distinction between GWAS using AILs and GWAS using humans. The ability to identify QTLs and eQTLs depends on genetic diversity, LD decay, and MAFs within the sample, which affect power by increasing or reducing the multiple testing burden. Compared to unrelated human populations, AILs have fewer SNPs, higher levels of LD, and higher MAFs (Fig. 1), all of which enhance power. We expected the increase in power conferred by an AIL to be less dramatic for cis-eQTLs than for trans-eQTLs due to the lower number of SNPs tested. This is because unlike human populations, which segregate far more variants (and thus, many more potential eQTLs), genetic diversity in an AIL is limited to variants segregating in the founder strains. Still, the proportion of eQTLs that were significant in all three brain regions was smaller for trans-eQTLs versus cis-eQTLs (Supplementary Figure 10), consistent with a larger proportion of the trans-eQTLs being false positives. Conversely, this observation could be explained by a high rate of false negatives due to the burden of testing all SNPs (trans-eQTLs) versus testing only nearby SNPs (cis-eQTLs). Thus, in addition to integrating QTLs with eQTLs, we incorporated data about gene expression in other tissues[12,18], coding SNPs, mutant mice, and human genetic studies to parse among

the implicated genes. Although that was broadly similar to the approach we applied to CFW mice, the eQTL sample in this study was more than two times greater. Indeed, to our knowledge, this is the largest eQTL analysis that has been performed in outbred mice.

We also used PheWAS to identify pleiotropic effects of several loci identified in this study. In many cases, pleiotropy involved highly correlated traits such as body weight on different days or behavior at different time points within a single day (Supplementary Figures 13–14, Supplementary Figure 16; Supplementary Data 1). We also observed unexpected examples of pleiotropy, for example, between locomotor activity and gastrocnemius mass on chromosome 4 (Supplementary Figure 17) and between locomotor activity and the startle response on chromosome 12 (Supplementary Figure 18). We observed extensive pleiotropy on chromosome 17 at ~26–30 Mb (Supplementary Data 1). In the current study we found that this locus influenced saline- and methamphetamine-induced locomotor activity and startle response (Supplementary Figure 10), and this same region was implicated in anxiety-like behavior[15], contextual and conditioned fear[15], and startle response[14] in previous studies of LG x SM G34, suggesting that the locus has a broad impact on many behavioral traits. These results support the idea that pleiotropy is a pervasive feature in model organisms and provides further evidence of the replicability of the loci identified by this and prior GWAS.

Discoveries from human GWAS are often considered preliminary until they are replicated in an independent cohort. There were several associations identified in LG × SM G34 mice that we replicated in G50–56[10–12,14,18], however, other findings from G34 did not replicate in G50–56. Replication of GWAS findings is a complex issue in both human and model organism genetics. Failures to replicate prior results in the current study could be due to methodological differences, as previous studies used slightly different statistical models and used data from both $F_2$ and $F_{34}$ populations. Alternatively, the failure of some loci to replicate could imply that some of our prior findings, while real, did not explain as much of the variance as we estimated due to a winner's curse. Indeed, even if the effect sizes of the earlier findings were correct, we would still not expect to replicate all of them, since power to replicate is seldom 100%.

In model organisms, it is also possible to replicate an association by directly manipulating the implicated gene. We replicated one behavioral locus identified in this study using the criteria of both human and model organism genetics. We had identified an association with locomotor activity on chromosome 8 using G34 of this AIL[13], which was replicated in the present study (Fig. 4). In both cases, the SM allele was associated with the lower activity (Fig. 4c, d). We also identified a locus for PPI (76 dB) in this region (Fig. 4a; Supplementary Data 1, Supplementary Figure 2). The loci identified in both G34 and in G50–56 were small and contained just one gene: Csmd1 (Fig. 4b). In the present study, we also identified a cis-eQTL for Csmd1 in HIP (Supplementary Figure 11). Finally, we obtained Csmd1 mutant mice[51] that also exhibited altered locomotor activity (Fig. 4e). Thus, we have demonstrated replication both by performing an independent GWAS and by performing an experimental manipulation that recapitulates the phenotype.

In summary, we have shown that LG x SM AIL mice are a powerful multigenerational intercross population that can be used for GWAS. The simple genetic background of an AIL makes it an appealing alternative to other multigenerational mouse populations, such as the Diversity Outcross[52]. We have identified numerous genome-wide significant loci for a variety of biomedically significant phenotypes. We also made use of eQTL data to parse among the genes implicated in particular loci. In the case of Csmd1, we showed replication in a second mapping population

and by directly manipulating the implicated gene, which produced a similar phenotype.

## Methods

**Genetic background.** The LG and SM inbred mouse strains (*Mus musculus domesticus*) were independently selected for high and low body weight at 60 days[53]. The LG × SM AIL was derived from an $F_1$ intercross of SM females and LG males initiated by Dr. James Cheverud at Washington University in St. Louis[8]. Subsequent AIL generations were maintained using at least 65 breeder pairs selected by pseudo-random mating[54]. In 2006, we established an independent AIL colony using 140 G33 mice obtained from Dr. Cheverud (Jmc:LG,SM-G33). Since 2009, we have selected breeders using an R script that leverages pairwise kinship coefficients estimated from the AIL pedigree to select the most unrelated pairs while also attempting to minimize mean kinship among individuals in the incipient generation (the full pedigree is included in Supplementary Data 6 and a link to the R script is in the Supplementary Note 1). We maintained ~100 breeder pairs in G49–55 to produce the mice for this study. In each generation, we used one male and one female from each nuclear family for phenotyping and reserved up to three of their siblings for breeding the next generation.

**Phenotypes.** We subjected 1123 AIL mice (562 female, 561 male; Aap:LG,SM-G50–56) to a four-week battery of tests over the course of two years. 1063 of these mice (530 female, 533 male) had high-quality GBS data; therefore, we restricted our analysis to these individuals. This sample size was based on an analysis suggesting that 1000 mice would provide 80% power to detect associations explaining 3% of the phenotypic variance (Supplementary Figure 19). We measured CPP for 1 mg kg$^{-1}$ methamphetamine, locomotor behavior, PPI, startle, body weight, muscle mass, bone length, and other related traits (Supplementary Data 2). We tested mice during the light phase of a 12:12 h light–dark cycle in 22 batches comprised of 24–71 individuals (median = 53.5). Median age was 54 days (mean = 55.09, range = 35–101) at the start of testing and 83 days (mean = 84.4, range = 64–129) at death. Mice were housed in same-sex cages, with 2–4 mice per cage. Standard lab chow and water were available ad libitum, except during testing. Testing was performed during the light phase, starting one hour after lights on and ending one hour before lights off. No environmental enrichment was provided. All procedures were approved by the Institutional Animal Care and Use Committee at the University of Chicago. Traits are summarized briefly below; detailed descriptions are provided in the Supplementary Methods.

CPP and locomotor behavior: CPP is an associative learning paradigm that has been used to measure the motivational properties of drugs in humans[55] and rodents[43]. We defined CPP as the number of seconds spent in a drug-associated environment relative to a neutral environment over the course of 30 min. The full procedure takes eight days, which we refer to as D1–D8. We measured baseline preference after administration of vehicle (0.9% saline, i.p.) on D1. On D2 and D4, mice were administered methamphetamine (1 mg kg$^{-1}$, i.p.) and restricted to one visually and tactilely distinct environment; on D3 and D5 mice were administered vehicle and restricted to the other, contrasting environment. No testing occurs on D6 and D7. On D8, mice were allowed to choose between the two environments after administration of vehicle; we measured CPP at this time. Other variables measured during the CPP test include the distance traveled (cm) on all testing days, the number of side changes on D1 and D8, and locomotor sensitization to methamphetamine (the increase in activity on D4 relative to D2). We measured CPP and locomotor traits across six five-minute intervals and summed them to generate a total phenotype for each day.

PPI and startle: PPI is the reduction of the acoustic startle response when a loud noise is immediately preceded by a low decibel (dB) prepulse[56]. PPI and startle are measured across multiple trials that occur over four consecutive blocks of time[14]. The primary startle trait is the mean startle amplitude across all pulse-alone trials in blocks 1–4. Habituation to startle is the difference between the mean startle response at the start of the test (block 1) and the end of the test (block 4). PPI, which we measured at three prepulse intensities (3, 6, and 12 dB above 70 dB background noise), is the mean startle response during pulse-alone trials in blocks 2–3 normalized by the mean startle response during prepulse trials in blocks 2–3. Mice that exhibited a startle response in the absence of a pulse were excluded from GWAS, as were mice that did *not* exhibit a startle response during the first block of startle pulses (Supplementary Methods, Supplementary Figure 20).

Physiological traits: We measured body weight (g) on each testing day and at the time of death. One week after PPI, we measured blood glucose levels (mg/dL) after a four-hour fast. One week after glucose testing, we killed the mice, and measured tail length (cm from base to tip of the tail). We stored spleens in a 1.5 mL solution of 0.9% saline at −80 °C until DNA extraction. We removed the left hind limb of each mouse just below the pelvis; hind limbs were stored at −80 °C. Frozen hind limbs were phenotyped in the laboratory of Dr. Arimantas Lionikas at the University of Aberdeen. Phenotyped muscles include two dorsiflexors, TA and EDL, and three plantar flexors: gastrocnemius, plantaris, and soleus. We isolated individual muscles under a dissection microscope and weighed them to 0.1 mg precision on a Pioneer balance (Ohaus, Parsippany, NJ, USA). After removing soft tissue from the length of tibia, we measured its length to 0.01 mm precision with a Z22855 digital caliper (OWIM GmbH & Co., Neckarsulm, GER).

Brain tissue: We collected HIP, PFC, and STR for RNA-seq from the brain of one mouse per cage. This allowed us to dissect each brain within five minutes of removing a cage from the colony room (rapid tissue collection was intended to limit stress-induced changes in gene expression). We preselected brain donors to prevent biased sampling of docile (easily caught) mice and to avoid sampling full siblings, which would reduce our power to detect eQTLs. Intact brains were extracted and submerged in chilled RNALater (Ambion, Carlsbad, CA, USA) for one minute before dissection. Individual tissues were stored separately in chilled 0.5-mL tubes of RNALater. All brain tissue was dissected by the same experimenter and subsequently stored at −80 °C until extraction.

**GBS variant calling and imputation.** GBS is a reduced-representation genotyping method[41,57] that we have adapted for use in mice and rats[5,42]. We extracted DNA from spleen using a standard salting-out protocol and prepared GBS libraries by digesting DNA with the restriction enzyme *PstI*. We sequenced 24 uniquely barcoded samples per lane of an Illumina HiSeq 2500 using single-end, 100 bp reads. We aligned 1110 GBS libraries to the mm10 reference genome before using GATK[58] to realign reads around known indels in LG and SM[21] (Supplementary Methods). We obtained an average of 3.2 million reads per sample. We discarded 32 samples with <1 million reads aligned to the main chromosome contigs (1–19, X, Y) or with a primary alignment rate <77% (i.e., three s.d. below the mean of 97.4%; Supplementary Figure 21).

We used ANGSD[59] to obtain genotype likelihoods for the remaining 1078 mice and used Beagle[60,61] for variant calling, which we performed in two stages. We used first-pass variant calls as input for IBDLD[62,63], which we used to estimate kinship coefficients for the mice in our sample. Because our sample contained opposite-sex siblings, we were able to identify and resolve sample mix-ups by comparing genetic kinship estimates to kinship estimated from the LG × SM pedigree (Supplementary Data 6, Supplementary Note 1). In addition, we re-genotyped 24 mice on the GigaMUGA[20] to evaluate GBS variant calls (Supplementary Table 1 lists concordance rates at various stages of our pipeline; see Supplementary Methods for details).

After identifying and correcting sample mix-ups, we discarded 15 samples whose identities could not be resolved (Supplementary Methods). Next, we used Beagle[60,61], in conjunction with LG and SM haplotypes obtained from whole-genome sequencing data[21] to impute 4.3 million additional SNPs into the final sample of 1063 mice. We excluded X chromosome SNPs to avoid potential problems with genotyping accuracy, statistical power, and other complications that have been discussed elsewhere[64]. We removed SNPs with low MAFs (<0.1), SNPs with Hardy–Weinberg Equilibrium (HWE) violations ($p \leq 7.62 \times 10^{-6}$, Chi-squared test), determined from gene-dropping simulations as described in the Supplementary Methods), and SNPs with low imputation quality (dosage $r^2$, $DR^2 < 0.9$). We then pruned variants in high LD ($r^2 > 0.95$) to obtain the 523,028 SNPs that we used for GWAS.

**LD Decay.** We used PLINK[65] to calculate $r^2$ for all pairs of autosomal GWAS SNPs typed in G50–56 (parameters are listed in Supplementary Note 1). We repeated the procedure for 3,054 SNPs that were genotyped in G34 mice[13]. Next, we randomly sampled $r^2$ values calculated for ~40,000 SNP pairs from each population and used the data to visualize the rate of LD decay (Fig. 1c).

**GWAS.** We used a leave one chromosome out LMM (LOCO-LMM) implemented in GEMMA[27] to perform GWAS. An LMM accounts for relatedness by modeling the covariance between phenotypes and genotypes as a random, polygenic effect, which we also refer to as a genetic relationship matrix (GRM). Power to detect associations is reduced when the locus being tested is also included in the GRM because the effect of the locus is represented in both the fixed and random terms[13,24]. To address this issue, we calculated 19 separate GRMs, each one excluding a different chromosome. When testing SNPs on a given chromosome, we used the GRM that did not include markers from that chromosome as the polygenic effect in the model. Fixed covariates for each trait are listed in Supplementary Data 2.

We used a permutation-based approach implemented in MultiTrans[66] and SLIDE[67] to obtain a genome-wide significance threshold that accounts for LD between nearby markers (Supplementary Methods). We obtained a significance threshold of $p = 8.06 \times 10^{-6}$ ($\alpha = 0.05$) from 2.5 million samplings. Because the phenotypic data were quantile-normalized, we applied the same threshold to all traits. We converted p-values to LOD scores and used a 1.5-LOD support interval to approximate a critical region around each associated region, which enabled us to systematically identify overlap with eQTLs.

**Trait correlations and heritability.** We calculated the Pearson correlation coefficients and their p-values for all pairs of traits measured in this study. Only mice with non-missing data for both traits in a pair were used to calculate the correlation. We also decomposed the trait covariance into genetic and environmental covariances using a multivariate LOCO-LMM in GEMMA[68]. This allowed us to evaluate the contribution of genetic and environmental factors to each pair of traits (Supplementary Figure 5, Supplementary Data 3).

We estimated the proportion of phenotypic variance explained by the additive effects of 523,028 LD-pruned SNPs using the restricted maximum likelihood algorithm in GEMMA[27]. Specifically, we ran a second genome-wide scan for each trait, this time dropping the fixed effect of dosage and including the complete GRM estimated from SNPs on all 19 autosomes. We refer to this estimate as SNP heritability $\left(\hat{h}_{SNP}^2\right)$. For comparison, we also estimated heritability using pedigree data $\left(\hat{h}_{Ped}^2\right)$. We used the same method described above, only the GRM was replaced with a kinship matrix estimated from the AIL pedigree (Supplementary Data 2, Supplementary Note 1, Supplementary Data 6). To estimate effect sizes for trait-associated SNPs, we repeated the procedure for $\hat{h}_{SNP}^2$ using dosage at the most significant SNP as a covariate for each trait (Supplementary Data 1). We interpreted the difference between the two estimates as the effect size of that locus.

**RNA-sequencing and quality control**. We extracted RNA from HIP, PFC, and STR using a standard phenol-chloroform procedure and prepared cDNA libraries from 741 samples with RNA integrity scores ≥8.0 (265 HIP; 240 PFC; 236 STR)[69] as measured on a Bioanalyzer (Agilent, Wilmington, DE, USA). We used Quant-iT kits to quantify RNA (Ribogreen) and cDNA (Picogreen; Fisher Scientific, Pittsburgh, PA, USA). Barcoded sequencing libraries were prepared with the TruSeq RNA Kit (Illumina, San Diego, USA), pooled in sets of 24, and sequenced on two lanes of an Illumina HiSeq 2500 using 100 bp, single-end reads.

Because mapping quality tends to be higher for reads that closely match the reference genome[70], read mapping in an AIL may be biased toward the reference strain (C57BL/6J)[71]. We addressed this concern by aligning RNA-seq reads to custom genomes created from LG and SM using whole-genome sequence data[21]. We used default parameters in HISAT[72] for alignment and GenomicAlignments[73] for assembly, assigning each read to a gene as defined by Ensembl (*Mus_musculus*. GRCm38.85)[74]. We required that each read overlap one unique disjoint region of the gene. If a read contained a region overlapping multiple genes, genes were split into disjoint intervals, and any shared regions between them were hidden. If the read overlapped one of the remaining intervals, it was assigned to the gene that the interval originated from; otherwise, it was discarded. Next, we reassigned the mapping position and CIGAR strings for each read to match mm10 genome coordinates and combined the LG and SM alignment files for each sample by choosing the best mapping. Only uniquely mapped reads were included in the final alignment files. We then used DESeq[75] to obtain normalized read counts for each gene in HIP, PFC, and STR. We excluded genes detected in <95% of samples within each tissue. We retained a total of 16,533 genes in HIP, 16,249 genes in PFC, and 16,860 genes in STR.

We also excluded 30 samples with <5 M mapped reads or with an alignment rate <91.48% (i.e., less than 1 s.d. below the mean number of reads or the mean alignment rate across all samples and tissues; Supplementary Figure 22). We merged expression data from HIP, PFC, and STR and plotted the first two principal components (PCs) of the data to identify potential tissue swaps. Most samples clustered into distinct groups based on tissue. We reassigned 12 mismatched samples to new tissues and removed 35 apparently contaminated samples that did not cluster with the rest of the data (Supplementary Figure 23). We also used agreement among GBS genotypes and genotypes called from RNA-seq data in the same individuals to identify and resolve mixed-up samples, as detailed in the Supplementary Methods. We discarded 108 sample mix-ups that we were not able to resolve, 29 samples with low-quality GBS data, and 12 outliers (details are provided in the Supplementary Methods). A total of 208 HIP, 185 PFC, and 169 STR samples were retained for further analyses.

**eQTL mapping**. Prior to eQTL mapping, we quantile-normalized gene expression data and used principal components analysis to remove the effects of unknown confounding variables[76]. For each tissue, we calculated the first 100 PCs of the gene expression matrix. We quantile-normalized PCs and used GEMMA[27] to test for association with SNPs using sex and batch as covariates. We evaluated significance with the same permutation-based threshold used for GWAS. We retained PCs that showed evidence of association with a SNP in order to avoid removing *trans*-eQTL effects. We then used linear regression to remove the effects of the remaining PCs (Supplementary Figure 24) and quantile-normalized the residuals.

We then mapped *cis*- and *trans*-eQTLs using a LOCO-LMM implemented in GEMMA[27]. We considered intergenic SNPs and SNPs 1 Mb upstream or downstream of the gene as potential *cis*-eQTLs and excluded 2143 genes that had no SNPs within their *cis*-regions. We used eigenMT[77] to obtain a gene-based *p*-value adjusted for the number of independent SNPs in each *cis* region. We declared *cis*-eQTLs significant at an FDR < 0.05 (Supplementary Data 4).

SNPs on chromosomes that did not contain the gene being tested were considered potential *trans*-eQTLs. We determined significance thresholds for *trans*-eQTLs by permuting the data 1000 times. Since expression data were quantile-normalized, we permuted one randomly chosen gene per tissue. The significance threshold for *trans*-eQTLs was $p = 8.68 \times 10^{-6}$ in STR, $p = 9.01 \times 10^{-6}$ in HIP, and $p = 1.04 \times 10^{-5}$ in PFC ($\alpha = 0.05$). We used all SNPs for permutation; therefore, we expect these thresholds to be slightly conservative. We define master eQTLs as 5 Mb regions that contain ten or more *trans*-eQTLs. To identify master eQTLs, we divided chromosomes into 5 Mb bins and assigned each *trans*-eQTL gene to the bin containing its most significant eQTL SNP.

**Csmd1 mutant mice**. *Csmd1* mutants were created by Lexicon Genetics by inserting a Neomycin cassette into the first exon of *Csmd1* using embryonic stem cells derived from 129S5 mice[78]. The mice we used were the result of a C57BL/6×129S5 intercross designated B6;129S5-*Csmd1*[tm1Lex]/Mmucd (the exact C57BL/6 substrain is unknown). We bred heterozygous males and females and tested littermate offspring to account for their mixed genetic background. *Csmd1* spans 1.6 Mb and has 70 exons. Its four major transcripts, termed *Csmd1-1* to *Csmd1-4*, are expressed in the central nervous system[51]. Distler et al. (ref. [51]) demonstrated that *Csmd1* homozygous mutant mice express <30% of wild-type *Csmd1* levels in the brain, and heterozygous mice show a 54% reduction in *Csmd1* expression. Residual expression of *Csmd1* in homozygous mutant mice is derived from *Csmd1-4*, the only transcript that does not include the first exon. We analyzed locomotor behavior on two days following a saline injection in 31 wild-type, 59 heterozygous, and 48 mutant mice.

**Code availability**. Example code for running the analyses in this study are provided in Supplementary Note 1. We also list the software used for these analyses (including version numbers) in the Supplementary Methods.

## Data availability

LG × SM G50–56 genotypes (hard calls and dosages), phenotypes (raw and quantile-normalized), and gene expression data (log2-normalized read counts and quantile-normalized data with PCs regressed out) are freely and publicly available on http://palmerlab.org/protocols-data/ and on http://genenetwork.org/ (Study ID 272). GeneNetwork accession codes are GN844–846 for RNAseq data (log2-normalized counts) and GN653 for phenotype data (raw), an accession code for genotype data is pending. All other relevant data is available upon request. Mouse phenotype and genotype data plotted in Fig. 4 is provided as a Source Data File.

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

## Acknowledgements

We are grateful to Heather Lawson at Washington University in St. Louis for providing LG and SM genome sequences. We thank the Gilad Lab and Functional Genomics Facility at the University of Chicago for generating DNA- and RNA-seq data. We wish to acknowledge outstanding technical assistance from Apurva Chitre at UCSD and Mike Jarsulic at the Biological Sciences Division Center for Research Informatics at the University of Chicago. We thank Clarissa Parker, John Novembre, Graham McVicker, Joe Davis, Peter Carbonetto and Shyam Gopalakrishnan for advice, training, and mentorship. Our work was funded by NIDA (AAP: R01DA021336) and NIAMS (AL: R01AR056280). We received additional support from NIGMS (NMG: T32GM007197; MGD: T32GM07281), NIDA (NMG: F31DA03635803), NHGRI (MA: R01 HG002899), and the IMS Elphinstone Scholarship at the University of Aberdeen (AIHC). The content is solely the responsibility of the authors and does not necessarily represent the official views of the NIH.

## Author contributions

N.M.G. maintained the AIL colony, phenotyped the mice, prepared RNA-sequencing libraries, and performed QTL/eQTL analysis under the supervision of A.A.P. and M.A. A.A.P. and M.A. also provided computational resources for the analyses in this paper. J.S. prepared RNA-sequencing data for eQTL mapping under supervision of S.C. C.L.S. assisted with colony maintenance, tissue collection, RNA extraction, and GBS. M.G.D. performed experiments in mutant mice. A.L., J.S.G., and A.I.H.C. collected skeletal muscle and bone phenotypes and advised on the analyses of these traits. N.M.G. co-wrote the manuscript with A.A.P., who designed the study.

## Additional information

**Competing interests:** The authors declare no competing interests.

