## [Peer Review File · Nature Communications]

Reviewers' comments:

Reviewer #1 (Remarks to the Author):

This manuscript describes a comprehensive association analysis of the LGxSM AIL mice. In particular, the authors performed GWAS on 50 traits in ~1000 mice and identified 126 genome-wide associations. They performed eQTL mapping analysis in three brain regions from ~200 mice and identified thousands of cis-eQTLs and trans-eQTLs. Importantly, the authors used a mutant mice line to further replicate one identified candidate gene. Overall, this is a well designed study with solid results that will have a long lasting impact in the field. The manuscript is also very well written. I only have a few comments which hopefully could improve the quality of the paper further:

1. One thing that was a bit surprising to me was the relatively large number of trans-eQTLs detected (i.e. ~1500 trans-eQTLs vs ~7000 cis-eQTLs). I am used to seeing only a few trans-eQTLs from human eQTL mapping studies with similar sample sizes. It would be useful to examine whether the nearby genes for these trans-SNPs tend to be transcription factors or not. It would be helpful if the authors could provide some discussion on the potential reasons that may facilitate the detection of trans-eQTL (vs cis-eQTLs) in mice studies. From Figure S4, it also seems that trans-eQTLs are more tissue specific than cis-eQTLs. It would be also nice to mention this finding and briefly discuss this finding in the results or discussion.

2. For heritability estimation, the authors have primarily focused on examining the proportion of phenotypic variance due to additive effects. Various previous studies have shown that epistasis may be an important contributor to phenotypic variegation in many model organisms. The large sample size and relatively large number of traits used in the present study may allow us to accurately estimate the epistatic contribution to the genetic architecture of different traits. I was wondering if it is possible to run a multiple variance component model to estimate the epistasis contribution for at least a few traits here. For example, the authors could follow the Supplementary Figure S3 in Crawford et al., 2018 (<https://www.tandfonline.com/doi/full/10.1080/01621459.2017.1361830?scroll=top&needAccess=true>) and use GEMMA to partition the phenotypic variance into multiple variance components that include an additive component and an epistasis component. If the analysis is too complicated, then providing some discussion on epistasis would be helpful.

3. In addition to estimating the phenotypic covariance in Figure S13, it would be nice to decompose the phenotypic covariance into two parts: a genetic covariance part and an environmental covariance part. For example, for each pair of traits in turn, one could use the mvLMM option in GEMMA to estimate these values. Such decomposition would allow us to understand whether the phenotypic covariance between pairs of traits are mostly due to genetics or environments.

Reviewer #2 (Remarks to the Author):

Gonzales et al. submitted an interesting manuscript dealing with genome wide association analysis making use of the mouse advanced intercross line (AIL). The authors outline the advantages of the LG/J x SM/J AIL for the study of traits that could be relevant for metabolic and psychiatric disorders. They briefly describe the research strategy in the introduction section highlighting the history over 15+ years after initiating AIL. Among other advantages compared to classical F2 crosses, the authors argue that AIL provides considerably higher mapping resolution. Using over 1000 mice from generations 50-56 of AIL, they identified 126 genome-wide associations for 50 traits. Among numerous others, these traits included locomotor activity, PPI, and body size but also metabolic parameters like glucose and morphometric parameters. Hippocampus, striatum, and prefrontal cortex

of about 200 mice were then used for RNA-seq to identify eQTLs that could be used for the interpretation of the discovered associations. Several examples are then discussed in a bit random way to highlight the potentials of this approach. As a proof-of-principle, a top candidate for a highly significant association, *Csmd1* was further verified by using a genetically engineered mouse model. The authors conclude, that *Csmd1* mutant mice recapitulate the phenotype association detected by AIL GWAS. Finally, the authors state that all data and statistical tools are publicly available on a project respectively institute website.

Overall, I believe the presented research is a creditable and valuable contribution to decipher mammalian gene functions. The paper is well written but to some degree the balance between rather methodological considerations and a genuine scientific question that was investigated was missed in my view. I wonder what was the major outcome of the study? 126 (new??) associations to the selected traits? – Or the confirmation of just one highly significant association between a candidate gene and locomotor activity that was verified in a mutant mouse? Or was it the seemingly successful implementation of (new?) statistical approaches.

To me, the rationale which traits were selected and why was not clear but confusing, especially as some traits even consisted of a behavioral response to drug treatment. I was impressed by the high number of associations found for morphometric traits which I assume are difficult to measure with sufficiently high precision (especially involving different people as experimenter). Here, I did not quite understand why muscle weight or bone length associations were pleiotropic with body weight associations (line 208) – a bigger mouse has higher muscle mass and maybe even longer bones?

In general, it was planned to study traits that are relevant for human metabolic and psychiatric disorders. However, only little support is given that the findings really have translational potential.

Finally, in line 298 the authors state that the primary goal was to identify the genes that are responsible for the loci implicated in behavioral and physiological traits, and “We were particularly interested in uncovering genetic factors that influence conditioned place preference...” – this specific goal was not clear to me until it was stated in the middle of the discussion.

Reviewer #3 (Remarks to the Author):

The authors report analysis of an advanced intercross line of mice for mapping of a large number of behavioral and metabolic traits. The authors also examined gene expression in several brain regions of a subset of the mice to identify local and trans-acting expression quantitative trait loci (eQTL). Altogether 126 genome-wide significant loci for about 50 traits were identified. One candidate gene was examined using gene targeted mice in order to validate the approach.

This is a thorough, well written manuscript that will be a valuable resource for understanding the genetic architecture of mammalian behavior. I identified some points that require clarification.

1. The heritabilities of certain traits as shown in Fig 3B are surprisingly low, given that this is a mouse study. For example, in a similar mouse study our lab estimated that body weight had a heritability of about 70-80%. The authors might comment.

2. I suspect that many of the trans eQTL are false positives, although those affecting multiple genes

look promising. Can the authors perform some sort of replication analysis (for example, dividing the mouse population in two) to address the issue?

3. A discussion of significance thresholds for mapping would be relevant. Since the authors examined 50 traits was that incorporated into the threshold calculations?

Reviewer #1

One thing that was a bit surprising to me was the relatively large number of trans-eQTLs detected (i.e. ~1500 trans-eQTLs vs ~7000 cis-eQTLs). I am used to seeing only a few trans-eQTLs from human eQTL mapping studies with similar sample sizes.

We appreciate this insightful comment. There are several reasons why the power to detect trans-eQTLs in our population is greater than in a human study of similar size. First, LD is more extensive in this population (Fig. 1c), meaning that the effective number of tests per GWAS is lower. Second, the high MAFs maintained in the AIL (Fig. 1b) mean that the tests are better powered. Third, the ability to control environmental variables (e.g. diet and housing conditions) during the lifetime of these mice may have enhanced our power to detect genetic effects on gene expression, compared to human studies in which there are numerous sources of environmental heterogeneity. Finally, the quality and consistency of our tissue is better than could be expected in a human study because the tissues are collected immediately after sacrifice by the same technician. All of these factors contributed to the large number of genome-wide significant eQTL that we observed. We have now reworked our presentation of these issues in the discussion.

However, we also should have been more explicit in acknowledging that a fraction of the “significant *trans*-eQTLs” are likely to be false positives. These *trans*-eQTLs were genome-wide significant at $\alpha=0.05$ after correction for the individual genome scan, but before attempting any correction for the number of genes tested. Given that $\alpha=0.05$, we expected that 5% of genes for which the null hypothesis was correct would be declared “significant” by this standard. Note that we were able to correct for the number of genes tested in the *cis*-eQTL analysis; hence, our abstract could have incorrectly given the impression that we found more *trans*-eQTLs than *cis*-eQTLs. We have now revised our paper at several points throughout the text to clarify and emphasize this critical point:

1. To avoid presenting *trans*-eQTL results that we acknowledge are a mixture of true and false positives, we have removed the number of *cis*- and *trans*-eQTLs from the abstract. The relevant sentences now read: *“Here we use 1,063 AIL mice to identify 126 genome-wide significant associations for 50 traits relevant to human health and disease. We also identify thousands of cis- and trans-eQTLs in the hippocampus, striatum, and prefrontal cortex of ~200 mice.”*
2. We moved the old Fig. 4 (the Circos plot) to the supplement (now Supplementary Fig. 9) in order to avoid placing undue emphasis on our *trans*-eQTL results. We also added Supplementary Fig. 8; this QQ plot shows the observed and expected p-values for all genes and all SNPs tested in the genome-wide scans for each tissue. This figure clearly illustrates that there is true signal in our *trans*-eQTL analysis.
3. We included a statement clarifying the expected false positive rate of 5% in the results and added a paragraph on the issues of power and false positives to the discussion.
4. In making Supplementary Fig. 8, we discovered that several *trans*-eQTL result files were incomplete due to a computational error. We have since re-run the genome-wide scans for genes with incomplete results. We now report an additional 161 *trans*-eQTLs in HIP, 218 *trans*-eQTLs in PFC and 147 *trans*-eQTLs in STR.

It would be useful to examine whether the nearby genes for these trans-SNPs tend to be transcription factors or not.

We tested this idea by using g:Profiler to obtain a list of all genes within each master *trans*-eQTL and using TFCheckpoint, a repository of mammalian transcription factors, to identify transcription factors. *However, we found that transcription factors tended to cluster unevenly around the genome, confounding this approach.* LD

between SNPs within each master-eQTL also made it difficult to envision how to construct an enrichment query that would produce statistically meaningful results in this population. In the end, we felt this analysis did not improve the paper and so have not mentioned the issue.

It would be helpful if the authors could provide some discussion on the potential reasons that may facilitate the detection of trans-eQTL (vs cis-eQTLs) in mice studies. From Figure S4, it also seems that trans-eQTLs are more tissue specific than cis-eQTLs. It would be also nice to mention this finding and briefly discuss this finding in the results or discussion.

Figure S4 is now Supplementary Fig. 6.

The differences in tissue specificity between *cis*- and *trans*-eQTLs likely reflect differences in both the number of false positives and the number of false negatives. False positives would not replicate across tissues. The apparently higher tissue specificity for *trans*-eQTLs likely reflects a greater proportion of false positive results in the *trans*-eQTLs, which, unlike the *cis*-eQTLs, were not corrected for the number of genes tested (see above). Reciprocally, in some cases we might have correctly identified a *trans*-eQTL in one tissue but failed to identify it in the other two due to a type 2 error. These two factors may be more important than true biological differences in the tissue specificity of *cis*- versus *trans*-eQTLs. We did not state this as a conclusion because it is difficult to know how many of the *trans*-eQTLs identified in this study are false positives. Likewise, it is difficult to know how many are false negatives. The tissues we examined had different sample sizes and contained different individuals (96 mice were sequenced in all three tissues); therefore, we expected the power to detect eQTLs to vary between tissues.

In response to this reviewer's question, we now mention the tissue specificity of the *trans*-eQTLs as being consistent with a higher rate of false positive results. We also added a note about detecting eQTLs in humans vs. mice to the discussion.

For heritability estimation, the authors have primarily focused on examining the proportion of phenotypic variance due to additive effects. Various previous studies have shown that epistasis may be an important contributor to phenotypic variegation in many model organisms. The large sample size and relatively large number of traits used in the present study may allow us to accurately estimate the epistatic contribution to the genetic architecture of different traits. I was wondering if it is possible to run a multiple variance component model to estimate the epistasis contribution for at least a few traits here. For example, the authors could follow the Supplementary Figure S3 in Crawford et al., 2018 and use GEMMA to partition the phenotypic variance into multiple variance components that include an additive component and an epistasis component. If the analysis is too complicated, then providing some discussion on epistasis would be helpful.

We appreciate the reviewer's comment; we have explored various ways to use these data to gain insights into the role of epistasis. Indeed, we have an ongoing project with another group that, while distinct from the approach taken by Crawford *et al.*, may provide some novel insights. We are also making the current data publically available and very much hope that this dataset will be useful for innovative analyses like the one suggested by this reviewer. After looking into this specific suggestion, we have decided not to pursue such an analysis, but do now mention the potential importance of epistasis for heritability in a newly added section of the discussion that addresses a comment by Reviewer #3.

In addition to estimating the phenotypic covariance in Figure S13, it would be nice to decompose the phenotypic covariance into two parts: a genetic covariance part and an environmental covariance part. For example, for each pair of traits in turn, one could use the mvLMM option in GEMMA to estimate these values. Such decomposition would allow us to understand whether the phenotypic covariance between pairs of traits are mostly due to genetics or environments.

This was great idea. We have performed these analyses and added the results to Supplementary Table 3. Since there are many pairwise comparisons, we also summarized the results in Supplementary Fig. 5. Also, Figure S13 is now Supplementary Fig. 4.

Reviewer #2

The paper is well written but to some degree the balance between rather methodological considerations and a genuine scientific question that was investigated was missed in my view. I wonder what was the major outcome of the study? 126 (new??) associations to the selected traits? – Or the confirmation of just one highly significant association between a candidate gene and locomotor activity that was verified in a mutant mouse? Or was it the seemingly successful implementation of (new?) statistical approaches.

The reviewer has correctly highlighted many of the major outcomes of our paper. We have made revisions to clarify these multiple significant outcomes and to highlight them in the narrative structure. We also wish to highlight the surprising (to us anyway) failure of CPP to show heritability and the utility of AIL mice relative to other mapping populations (e.g. the collaborative cross and diversity outcross) that have received more attention in the last few years. The abstract, introduction and discussion have all be changed to more clearly communicate the multiple outcomes of this paper.

To me, the rationale which traits were selected and why was not clear but confusing, especially as some traits even consisted of a behavioral response to drug treatment.

We were primarily interested in behavior and have tried to make this clear in this revised version. We also tested a variety of traits that our lab was familiar with and that we knew to be heritable in this AIL. Body weight and glucose levels were selected for these reasons and because they are easy to measure. We established collaborations with researchers interested in physiological phenotypes in order to produce more data and to get a better overall impression of the suitability of this AIL for GWAS and the behavior of polygenic traits in a system with only two founders. Thus, it was our intention to measure a variety of traits. We have revised the introduction and first paragraph of the discussion to communicate these ideas.

I was impressed by the high number of associations found for morphometric traits which I assume are difficult to measure with sufficiently high precision (especially involving different people as experimenter).

Great point. The multiple associations for body weight, muscle weight and bone size may reflect the fact the founders of this AIL (SM/J and LG/J) were selectively bred for small (SM) and large (LG) body size prior to inbreeding. We also measured body weight at multiple time points, which further contributes to the large number of associations. We have clarified this in the introduction and results to highlight these issues (and the ones below).

Because we were also concerned about involving different experimenters in our study, we included experimenter as a covariate in GWAS for muscle and bone phenotypes to account for experimenter-induced variability.

Here, I did not quite understand why muscle weight or bone length associations were pleiotropic with body weight associations (line 208) – a bigger mouse has higher muscle mass and maybe even longer bones?

Yes, body size is partially a function of muscle weight and bone size, and some of the loci that influence body weight do so via their ability to change parameters like muscle weight or bone size. This was not a forgone conclusion, since other measurements (fat pads, organ weight) could also change body weight. Since it is difficult to know whether or not the genetic factors influencing these different traits are acting independently, we use the term pleiotropy instead of claiming a causal relationship. We now explicitly state this when we introduce the concept of pleiotropy in the results section on physiological traits.

In general, it was planned to study traits that are relevant for human metabolic and psychiatric disorders. However, only little support is given that the findings really have translational potential.

In this paper we used established behavioral and physiological phenotypes, many of which have been developed and validated in terms of their clinical potential over the past several decades. We have provided more detailed descriptions of the phenotypes with additional citations in the Supplementary Note. That said, our paper is not primarily focused on explaining or defending the translational value of these traits and given the earlier comments by this reviewer, we did not want to further dilute the narrative by emphasizing these points in the main text.

Finally, in line 298 the authors state that the primary goal was to identify the genes that are responsible for the loci implicated in behavioral and physiological traits, and “We were particularly interested in uncovering genetic factors that influence conditioned place preference...” – this specific goal was not clear to me until it was stated in the middle of the discussion.

We appreciate this important comment and have revised the introduction to make this clearer.

Reviewer #3

The heritabilities of certain traits as shown in Fig 3B are surprisingly low, given that this is a mouse study. For example, in a similar mouse study our lab estimated that body weight had a heritability of about 70-80%. The authors might comment.

This is an important point that we failed to address in our initial submission. The discrepancy between heritability estimated from family studies and heritability estimated from unrelated individuals is well known in human genetics; family studies consistently report higher heritabilities across a variety of traits. The explanations for this include epistasis (see comments by Reviewer #1) and shared environmental factors in twin studies that result in higher phenotypic similarity among identical twins (and more generally among family members), making them difficult to disentangle from genetic effects. In mice and other model organisms, twin models are replaced by panels of inbred strains, which afford even greater control over the environment and can result in higher heritability estimates. We have added a paragraph in the discussion stating that these values are expected to be lower than those obtained using inbred strains.

I suspect that many of the trans eQTL are false positives, although those affecting multiple genes look promising. Can the authors perform some sort of replication analysis (for example, dividing the mouse population in two) to address the issue?

We agree that the *trans*-eQTL analysis is likely to include false positives, as described in our responses to Reviewer #1. We did not attempt to replicate by dividing the sample due to concerns about the loss of power in the smaller samples. The lack of replication across tissues for *trans*- compared to *cis*-eQTLs is now mentioned in the discussion as evidence of a higher false positive rate. We are making these data publically available; future work by our group or others might explore the *trans*-eQTLs further.

A discussion of significance thresholds for mapping would be relevant. Since the authors examined 50 traits was that incorporated into the threshold calculations?

We now discuss the issue of false positives for the behavioral, physiological and gene expression traits (see our comments to Reviewer #1). It is clear that we see more positive results than expected under the null for the behavioral, physiological and *cis*- and *trans*-eQTL results. We did not use an approach like a Bonferroni correction because it would be overly stringent since many of the traits are highly correlated with one another and because we do not believe (philosophically) that the null hypothesis (i.e. that no SNPs influence these traits) is true for most of these traits.

Reviewers' comments:

Reviewer #1 (Remarks to the Author):

My previous comments are well addressed and the paper is ready for publication.

Reviewer #2 (Remarks to the Author):

I carefully read the paper again. In my opinion all issues raised by the reviewers including my own comments were convincingly addressed.

Reviewer #3 (Remarks to the Author):

I am generally satisfied by the responses to my points as well as those of the other reviewers, but I feel that the discussion of trans eQTL is somewhat misleading. Based on previous studies I think the vast majority will be false positives.

I had asked the authors to divide the data in half and see if the results are consistent. They responded that they did not want to do that because would result in loss of power, which of course is true. But it might at least reveal outlier effects. What about checking for outliers, or dividing the data and asking if the associations go in the same direction. This is not a major point but it would be a simple thing to do.

We thank reviewers 1 and 2 for agreeing that all of their comments were addressed.

Reviewer 3 expressed **one remaining concern** about the *trans*-eQTL analysis; in particular, in their most recent comments (9-28-18) they said ***“I feel that the discussion of trans eQTL is somewhat misleading. Based on previous studies I think the vast majority will be false positives.”*** We agree.

In our responses to the first round of reviews, we indicated our agreement that many of the trans-eQTL results were false positives. Both Reviewer #1 and Review #3 commented on this topic, thus, much of our response was addressed to Reviewer #1, and may have been missed by Reviewer #3. In those responses, we stated:

- “...These *trans*-eQTLs were genome-wide significant at $\alpha=0.05$ after correction for the individual genome scan, but before attempting any correction for the number of genes tested. Given that $\alpha=0.05$, ***we expected that 5% of genes for which the null hypothesis was correct would be declared “significant” by this standard.*** Note that we were able to correct for the number of genes tested in the *cis*-eQTL analysis; hence, ***our abstract could have incorrectly given the impression that we found more trans-eQTLs than cis-eQTLs. We have now revised our paper at several points throughout the text to clarify and emphasize this critical point:***
 - “To avoid presenting *trans*-eQTL results that we acknowledge are a mixture of true and false positives, ***we have removed the number of cis- and trans-eQTLs from the abstract.*** The relevant sentences now read: “Here we use 1,063 AIL mice to identify 126 genome-wide significant associations for 50 traits relevant to human health and disease. We also identify thousands of *cis*- and *trans*-eQTLs in the hippocampus, striatum, and prefrontal cortex of ~200 mice.”
 - “***We moved the old Fig. 4 (the Circos plot) to the supplement*** (now Supplementary Fig. 9) in order to avoid placing undue emphasis on our *trans*-eQTL results. ***We also added Supplementary Fig. 8;*** this QQ plot shows the observed and expected p-values for all genes and all SNPs tested in the genome-wide scans for each tissue. This figure clearly illustrates that there is true signal in our *trans*-eQTL analysis.”
 - “***We included a statement clarifying the expected false positive rate of 5% in the results and added a paragraph on the issues of power and false positives to the discussion.***”
- “In response to this reviewer’s [Reviewer #1’s] question, ***we now mention [in the discussion] the tissue specificity of the trans-eQTLs as being consistent with a higher rate of false positive results.*** We also added a note about detecting eQTLs in humans vs. mice to the discussion.”

Therefore, the revised version of our manuscript very openly stated that many of the *trans*-eQTLs are false positives, as Reviewers #1 and #3 requested. However, we failed to reiterate these responses to Reviewer #3 directly.

Reviewer #3 had also asked us to perform an additional analysis in which we split the sample into two halves, to assess replication: “*I had asked the authors to divide the data in half and see if the results are consistent. They responded that they did not want to do that because would result in loss of power, which of course is true. But it might at least reveal outlier effects. What*

about checking for outliers, or dividing the data and asking if the associations go in the same direction. This is not a major point but it would be a simple thing to do.”

First of all, we found the reviewer's comments about outliers to be confusing. We have quantile normalized all expression data; thus outliers should not be a concern.

To address this reviewer's comments, we have now divided the data to determine if the associations were in the same direction (see Supplementary Information section 4.3, which is copied below). We then examined the effect size of each SNP that was called significant in the original analysis and compared the sign of the effect across the two subsets for each tissue. For hippocampus, 95.2% of effect sizes at significant *trans*-eQTLs were concordant among the random subsets (Pearson's $r^2=0.922$). For prefrontal cortex, effect size concordance was 84.8% (Pearson's $r^2=0.889$). For striatum, effect size concordance was 99.7% (Pearson's $r^2=0.923$).

We are happy to include Supplementary Information section 4.3 and a reference to it in the main text of our revised manuscript; however, we are concerned because the analysis does not resolve the question of how many *trans*-eQTL results are false positives. Indeed, it could be interpreted to indicate that most of them are true positives. This is because this approach does not estimate the study-wide false positive rate since the subsets are not independent from the discovery set. Thus, effect sizes are expected to be biased in a way that makes replication more likely.

We also considered a different approach in which one half would be used for discovery and the second for replication, however, since the discovery sample would be smaller, it would produce an even higher proportion of false positives, which would not elucidate the key question of what fraction of our reported *trans*-eQTLs were false positives. Thus, we have not taken this approach.

In summary, the manuscript is currently written to clearly indicate that many of the *trans*-eQTL results are false positives, which is the key concerns expressed by Reviewer #3. We are willing to include the additional analysis in Supplementary Information section 4.3, however, we are concerned that it is misleading and will give the impression that the majority of our *trans*-eQTL results are true positives. Thus, we somewhat prefer not to include this new analysis, but will leave it to the editor and Reviewer #3 to decide.

4.3. Significance of *trans*-eQTLs

These *trans*-eQTLs called “significant” in this study were genome-wide significant at $\alpha=0.05$ after correction for the individual genome scan, but before attempting any correction for the number of genes tested. Given that $\alpha=0.05$, we expected that 5% of genes for which the null hypothesis was correct would be declared “significant” by this standard. We performed an internal replication analysis of *trans*-eQTLs in HIP, PFC and STR by randomly splitting each sample in half and separately mapping eQTLs in the two subsets. We used the Wald test (`lmm -1` option in GEMMA), which yields effect size estimates (β) for each SNP. We then examined the effect size of each SNP that was called significant in the full sample, comparing the sign of the effect across the two subsets in each tissue. For HIP, 95.2% of effect sizes at significant *trans*-eQTLs were concordant among the random subsets (Pearson's $r^2=0.922$). For PFC, effect size concordance was 84.8% (Pearson's $r^2=0.889$). For STR, effect size concordance was 99.7% (Pearson's $r^2=0.923$). We note that this test does not estimate the study-wide false positive rate because the subsets are not independent of the discovery set. Thus, beta estimates are expected to be biased in a way that makes replication more likely.

REVIEWERS' COMMENTS:

Reviewer #3 (Remarks to the Author):

The authors have done a thorough job of addressing my final comments.